# Functionalized Wool as an Efficient and Sustainable Adsorbent for Removal of Zn(II) from an Aqueous Solution

**DOI:** 10.3390/ma13143208

**Published:** 2020-07-18

**Authors:** Marjana Simonič, Lidija Fras Zemljič

**Affiliations:** 1Laboratory for Water Biophysics and Membrane Processes, Faculty of Chemistry and Chemical Engineering, University of Maribor, Smetanova 17, SI-2000 Maribor, Slovenia; marjana.simonic@um.si; 2Laboratory for Characterization and Processing of Polymers, Faculty of Mechanical Engineering, University of Maribor, Smetanova 17, SI-2000 Maribor, Slovenia

**Keywords:** wool, chitosan, binding, sorption, Zn, thermodynamics

## Abstract

In this paper, the aim of the research was to obtain a highly efficient wool-based sorbent for the removal of zinc Zn(II) from wastewater. To increase the functional groups for metal binding, the wool was functionalized with chitosan. Chitosan has amino groups through which metals can be complexed easily to chelates. The physical and chemical modification of chitosan on wool was performed to analyze the influence of the coating bond on the final ability of the wool to remove metals. The presence of functional chitosan groups onto wool after adsorption was verified by attenuated total reflectance-Fourier transform infrared spectroscopy (ATR-FT-IR) spectra. The effective binding of chitosan to wool was also determined by potentiometric and polyelectrolyte titration methods. The latter titration was used to analyze the chitosan desorption. The main part of the study was the sorption of Zn(II) on natural and functionalized wool. The influence was investigated as a function of contact time, pH, metal ion concentration and temperature on the sorption process. The absorbent with the highest concentration of protonated amino groups (607.7 mmol/kg) and responding sorption capacity of 1.52 mg/g was obtained with wool physically modified by a macromolecular chitosan solution (1%) at pH = 7. Adsorption of Zn(II) onto pristine and modified wool corresponded to pseudo-second order kinetics (*R^2^* > 0.9884). The Langmuir model was found to be more suitable (*R^2^* > 0.9866) in comparison to the Freundlich model. The Zn(II) sorption process was spontaneous (∆*G* < 0) and exothermic (∆*H* < 0). The results found in this study are significant for escalating the possible use of wool modified with polysaccharide coatings as a sustainable source to improve or increase the metal sorption activity of wool.

## 1. Introduction

Recently, scientists have become increasingly interested in using wool as a natural fiber for potential biosorbents because of its abundance. Various techniques have been developed to improve the physical and mechanical properties of wool and its sorption capacity for a wide range of metals. Among them, it is of great interest to functionalize wool by polymers that have an affinity for metals with their functional groups (i.e., the amino group -NH_2_). However, due to the sustainability trend, the focus is on natural and biodegradable polysaccharides, including chitosan. Chitosan, with its amino groups, has a strong tendency to bind heavy metals on the basis of electron-donating nitrogen atoms [1].

The physical modification of wool with chitosan solutions was of particular interest in terms of improving the properties of wool by dyeing. Bendak et al. [2] determined whether the physical interactions between wool and chitosan were strong enough to improve the dyeing properties of wool. The wool samples were treated with chitosan solutions of different concentrations (0.1–1%) at different temperatures (70, 80 and 90 °C). With a higher concentration of chitosan, functionalized wool improved the uptake and intensity of color after dyeing. Additionally, Zhang et al. [3] found that physical modification of wool with chitosan had a beneficial effect on the final properties of wool (loss of strength, shrinkage, etc.). Here, the optimal conditions were achieved with the use of a chitosan solution at 6 g/L, pH = 5 and a temperature of 90 °C.

Xu [4] performed a chemical modification of chitosan on wool with environmentally friendly horseradish peroxidase to obtain its antimicrobial character. The reaction mechanism, the amount of binding and the properties of the chemically treated wool (wetting, shrinkage, antibacterial effect) were analyzed. A grafting efficiency of 2% to 12.6% was achieved with improved physical and mechanical properties, as well as positive changes in the hydrophilicity and antibacterial activity of the wool, together with better shrinkage resistance. A chemical modification with 1% and 2% (w/V%) chitosan solution over an anhydrous bridge with two anhydrides (succinic anhydride and phthalic anhydride) and solvents (dimethyl sulfoxide (DMSO) and N,N-dimethylformamide (DMF)) was shown in the study performed by Ranjbar-Mohammadi et al. [5]. The effect was investigated of anhydrides’ concentrations, the used solvent, pH, chitosan concentration and reaction time onto the adsorption affinity of chitosan to wool. It was proved that the grafting of chitosan to wool was due to established covalent bonds. The modified wool took over the antibacterial activity of chitosan, and also gained in shrinkage resistance during the washing process. Gawish et al. [6] performed the chemical modification of wool fibers by grafting the chitosan in the presence of citric acid as a crosslinking agent. This fiber modification improved the properties of wool, such as improved microbial resistance, dye absorption, tensile strength, significantly. The aim of the study by Periolatto et al. [7], was to achieve a surface modification of wool fibers with chitosan, which would improve the properties of the textile and its use. Through radical reactions (i.e., ultraviolet radiation), adsorption of 2% acidic chitosan solution onto wool was done. It has been shown that chitosan is useful in wool finishing as a shrinkage resistant agent, and as an agent to improve the dyeability of wool with the introduction of antimicrobial activity [8,9].

In addition, wool as a sorption substrate has attracted the attention of many scientists because of its high ability to remove metals to trace concentrations [10].

Zhang [11] removed various metals (Cr(III), Cu(II), Cd(II) and Pb(II)) with different biosorbents: chicken feathers, human hair, dog hair and pristine wool. The latter showed the highest ability to remove heavy metals. The influence of pH, sorbent concentration and contact time were examined on the sorption efficiency of pristine wool. With pristine wool, the sorption equilibrium was reached at 6 h, and the increased concentration of metals also increased the sorption efficiency. The optimum pH of the process was reached between pH values of 5 and 6. Moreover, the sorption kinetics, isotherms and Gibbs free energy were determined also. The time profile of wool sorption corresponded to the second order (R^2^ > 0.9985). By comparing Freundlich’s and Langmuir’s models, a better fit was achieved with the latter (R^2^ > 0.9999). Even Dakiky et al. [12] removed Cr(VI) with several sorbents: olive cake, wool, sawdust, pine needles, almond peels, cactus leaves and charcoal. They examined the influence of pH, contact time, metal concentration and the nature of the sorbent onto sorption efficiency. Under given conditions, the wool sorption equilibrium was reached after 1.5 h. The sorption of Zn(II) on unmodified wool fibers was investigated by Ahmed et al. [13]. By optimizing pH (7), contact time (2 h), metal ion concentration (50 mg/L) and sorbent concentration (70 g/L), 99.52% of Zn(II) removal efficiency was reached. Atef El-Sayed et al. [14] compared the sorption of Zn(II) and Cu(II) between untreated wool fibers, wool powder and oxidized wool powder, in order to optimize metal ion concentration, pH and contact time. Wool powder and oxidized wool powder achieved higher sorption capacity due to their bigger surface area and higher amount of available functional groups. It also indicated a higher binding capacity for Cu(II). Wool fibers took the longest time to reach sorption equilibrium (t > 2 h). The sorption of three metals—Cr(III), Cd(II) and Pb(II)—onto wool irradiated by electron beams was carried out by Hanzlíková et al. [15]. The effect was studied of the irradiated dose on the sorption efficiency of metal ions. Modified wool showed a higher sorption ability in comparison to untreated. Enkhzaya et al. [16] removed Cu(II), Pb(II) and Cd(II) with sheep’s wool, its excrement and fat. With 15 mL of a metal solution of an initial concentration of 10 mg/L, 0.05 g of wool achieved a sorption efficiency of more than 80%.

After reviewing the literature, we have found that sorption of Zn(II) or any other metal has not yet been carried out systematically with functionalized wool, physically or chemically modified with chitosan solutions. Zn(II) sorption was demonstrated sucesfully only by pristine wool and other biosorbents, but not on wool funcionalized by chitosan. Chitosan adsorbed onto wool was, until now, researched more as a finishing agent; however, it has not yet been investigated as an additive to improve wool sorption affinity.

We came to the realization that, although many studies exist on pristine wool [2,3] as a sorbent, there are still may challenging things to be understood in depth, such as assistance to exploit available natural products like wool to be functionalized further with biodegradable polymers; such as chitosan and; thus, upgraded in higher added value products as, for example, filters or sorbents for metal removal in waste water and, potentially, in thick sludges. The fundamental knowledge of functionalized wool properties and its influence on sorption behaviour needs to be understood for its real application. Especially important are sorbent functional groups, to which more attention needs to be paid, as shown in our work. It is assumed that these groups may act as the driving force for sorption.

The aim of the study was to obtain a highly efficient wool adsorbent for the removal of Zn(II) from waste water. The physical and chemical modification of chitosan onto wool was performed with the aim to introduce additional metal-binding places (i.e., functional groups). Both binding approaches were compared from the final sorption behavior. It is hypothesized that the main problem with the use of chitosan as a functional agent for wool is in its weak bonding to wool. This may be due to insufficient ionic interactions between the negatively charged carboxyl groups on the wool and the positively charged amino groups of chitosan, as well as weak Van der Waals forces and hydrogen bonds. Due to the coating instability, desorption is; therefore, possible, and may lead to a loss of functionality (i.e., metal sorption affinity). Thus, chitosan was also grafted onto wool chemically, and their characterisctics were compared versus physically modified fibers. To the best of the authors’ knowledge, no such complex and detailed study on the use of chitosan as a polysaccharide coating for the functionalisation of wool has been presented to date.

## 2. Materials and Methods

### 2.1. Materials

A standard Zn solution was prepared by dissolving 1g/L of Zn(II) using ZnSO_4_.7 H_2_O. For all the following sections this Zn solution was used to prepare initial solutions of Zn(II), except for the Atomic absorption spectroscopy (Varian, Inc., Palo Alto, CA, USA) calibration curve, where the working Zn(II) standard solution was prepared with concentration 10 mg/L by diluting the standard Zn-solution. The solutions were prepared with initial concentrations of 25, 12.5 and 6.25 mg/L for sorption studies.

#### 2.1.1. Wool Sample Preparation and Modification

Pristine sheep’s wool (purchased from Soven d.o.o., Selnica ob Dravi, Slovenia) was demineralized by rinsing in 0.001 M HCl and stirred for 30 min. Then it was washed to constant conductivity and dried overnight.

##### Physical Modification

For physical modification of wool samples, a 1% chitosan (Mw = 82,000 from Sigma Aldrich, Vienna, Austria, by 75–85% deacetylated degree) macromolecular solution was prepared at pH = 5 and 7 (adjusted with concentrated acetic acid or 0.1 M NaOH). The prepared solutions were stirred on a magnetic stirrer for 24 h at room temperature, and final pH was controlled as mentioned above. A total of 5 g of wool was functionalized to be inserted in both prepared chitosan macromolecular solutions, respectively, for 20 min, filtered further and then washed with deionized water. Finally, fibers were dried at a temperature of 70 °C for 30 min.

##### Chemical Grafting

Grafting of wool samples (0.6 g of wool sample) was done with the N-hydoxysucinimide (NHS) and 2-ethyl-3-(3-dimethyl-aminopropyl)carboimide (EDC) solution in the ratio 2:1 at pH = 5. The solution was stirred in an ice bath for 2 h. Chitosan (1%) was added drop wise. Then it was stirred for 48 h. Finally, the wool was filtered, washed with deionized water and dried at 70 °C for 30 min.

Samples were labelled as seen from Table 1. The samples are likewise shown in Figure 1.

### 2.2. Methods

#### 2.2.1. Wool Characterization

##### Attenuated Total Reflectance—Fourier Transform Infrared Spectroscopy (ATR-FTIR)

The wool samples were characterized by Fourier-transform infrared spectroscopy. During the analysis of the obtained spectra, characteristic peaks were resoluted, which indicate the presence of defined functional groups. The Perkin Elmer (Omega d.o.o., Ljubljana, Slovenia) FT-IR spectrophotometer was used as the analytical instrument, whereby the samples were placed in a compression diamond cell. The spectra for each sample were recorded 16-fold, the measured area was in the interval between 4000 and 650 cm^−1^, and 10 cm^−1^, at room temperature.

#### 2.2.2. Potentiometric Titration

The pH-dependent titration was carried out at pH = 2–11, using 0.1 M HCl and 0.1 M KOH aqueous solutions as titrants. A double burette instrument (Mettler T-70, Mettler Toledo, Ljubljana, Slovenia) was fitted out with a combined glass electrode (Mettler Toledo InLab Reach 225, Mettler Toledo, Ljubljana, Slovenia). All suspensions (1 g fibre/200 mL solution) for potentiometric titrations were prepared with ultrapure water with a very low carbonate content (<10^−6^ M), which was achieved by boiling and then cooling under a nitrogen atmosphere. Before the titration, the ionic strength of the wool suspension was adjusted to 0.1 M with 3 M KCl. The HCl-KOH blank titration was carried out under the identical circumstances as mentioned above. A detailed description regarding the determination of the charge calculations can be found in reference [17].

#### 2.2.3. Polyelectrolyte Titration

With desorption experiments it is conceivable to control the stability of chitosan applied to wool. Desorption was carried out by plunging 2 g of the wool sample in 100 mL of acidified water (pH = 3.6; adjusted with 0.1 M HCl), keeping the fibers at 37 °C for 2 h and shaking them thoroughly with a laboratory magnetic stirrer. After 2 h the wool samples were removed from the laboratory flasks and the supernatants of the remaining liquids were analyzed by polyelectrolyte titration. A Mettler Toledo DL 53 titrator (Mettler Toledo, Ljubljana. Slovenia) with a 10 mL burette was used to evaluate the chitosan desorption. The analyte consisted of 30 mL filtered solution (desorbed chitosan), 9 mL distilled water and a few drops of the indicator toluidine blue O (Sigma-Aldrich, Steinheim, Germany). Prior to titration, the pH value was adjusted to 3.6 with HCl to fully establish the protonation of chitosan. Every 5 to 10 s, 50 µL of the polyelectrolyte titrant polyethylene sulfonate sodium salt (PES-Na, c = 10 mM) were added. The absorbance was measured as potential change (mV) with a Mettler-Toledo Phototrode DP660 (Mettler Toledo, Ljubljana. Slovenia) at 660 nm. The NH_3_^+^ concentration was determined from the equivalent volume (V) of the added polyethylene sulfonate sodium salt presumptuous 100% stoichiometry. Three replicates were done for each sample.

#### 2.2.4. Sorption Experiments

##### Atomic Absorption Spectroscopy (AAS)

AAS was applied for measuring Zn(II) concentration, based on a calibration curve plot using a Perkin Elmer 3110 (Omega d.o.o., Ljubljana, Slovenia) spectrophotometer. Prior to analysis, the calibration curve of Zn(II) was prepared in a concentration range of 0.4 to 2.0 mg/L Zn(II) by diluting the Zn(II) standard solution (γ(Zn) = 1 g/L). Absorbances of the prepared Zn(II) standard solutions were measured at a wavelength of 213.9 nm. Each measurement was carried out in triplicate and the average results are stated. The standard deviation with this analytical technique was calculated to 10% of the measured value.

Kinetics was studied by the Langmuir model based on Equation (1):(1)$$\frac{C_{e}}{q_{e}}=\frac{1}{K_{L}q_{max}}+\frac{1}{q_{max}}c_{e}$$
where *q*_max_ is the maximum sorption capacity (mg/g), *K*_L_ is the Langmuir constant (L/mg), c_e_ is the equilibrium concentration of Zn(II) ions (mg/L) and *q*_e_ is the equilibrium capacity of Zn(II) ions (mg/g) [18].

A Freundlich model which proposed a heterogenic sorbent surface was also applied:(2)$$\log q_{e}=\log K_{F}+\frac{1}{n}\log c_{e}$$
where *K_F_* (mg^1−(1/n)^/gl^n−1^) and 1/*n* are Freundlich constants.

The Kinetical models chosen were the pseudo-first and pseudo-second order reactions. [19].

Equations (3) and (4) show both reactions:

(3)$$\mathrm{Pseudo}-\mathrm{first}:\log\left(q_{e}-q_{i}\right)=\log q_{e}-\frac{k_{1}}{2,303}t$$(4)$$\mathrm{Pseudo}-\mathrm{second}:\;\frac{t}{q_{i}}=\frac{1}{k_{2}q_{e}^{2}}+\frac{t}{q_{e}}$$
where *q_e_* = the equilibrium concentration of adsorbed ions (mg/g)

*q_i_* the concentration of adsorbed ions (mg/g) in time *t* (h)*k_1_* Pseudo-first order constant (h^−1^)*k_2_* pseudo-second order constant (g/mg h) [11,20]

The equilibrium of Zn(II) was measured at different temperatures—25, 35 and 45 °C. A total of 0.2 g of wool was added to the Zn(II) solution with a concentration of 12.5 mg/L. It was stirred for 24 h at 240 rpm. After filtration, AAS was used in order to determine Zn(II).

Gibbs free energy (∆*G⁰*) is connected to *K_c_°* as follows:(5)$$\Delta G^{{}^{\circ}}=-RTlnK_{c}^{{}^{\circ}}.\;$$
(6)$$\Delta G^{{}^{\circ}}=\Delta H^{{}^{\circ}}-T\Delta S^{{}^{\circ}}.\;$$

In addition, after the combination of Equations (5) and (6), we gain:(7)$$\ln\left(K_{c}\right)=\frac{\Delta S^{{}^{\circ}}}{R}-\frac{\Delta H^{{}^{\circ}}}{RT}$$

K_d_ = K_c_ = q_e_/c_e_ (L/g).

∆*H⁰* and ∆*S⁰*, were determined with plotting ln*(K_c_)* to *(1/T),* ∆*H⁰* was the slope and ∆*S⁰* the intercept. *R* equals 8.314 J/(mol.K) [21].

## 3. Results and Discussion

### 3.1. Characterization of Wool

#### 3.1.1. Chemical Composition of Functionalized Wool

The ATR-FTIR spectra of pristine wool and physical and chemically modified wool using chitosan are shown in Figure 2. Chitosan (H), as a reference, has characteristic signals over 3345.23 cm^−1^ with stretch vibrations of O–H and N–H, while the signal at 2865.07 cm^−1^ shows C–H stretch vibrations. Important signals were found at 1649.11 cm^−1^ (C=O bond—amide I) and 1585.68 cm^−1^ (N–H bond—amide II). The wide signal at 1026.65 cm^−1^ represents the asymmetric vibrations of C–O–C. The signals at 1373.56 cm^−1^ are vibrations of the C–H bond [22,23]. All the detected signals of chitosan origin present typical functional groups of deacetylated chitosan.

The wool sample has three main functional groups: –COOH, –NH_2_ and –OH [24]. The wide signal at 3277.77 cm^−1^ represents the vibrations of the N–H bond of primary amines, which is not charged, and –OH groups. Signals at 1629.29 and 1526.21 cm^−1^ represent –CONH_2_ (asparagine and glutamine) and –CONH– (peptide chain), respectively [14]. At wave number range 1450 to 1400 cm^−1^, characteristic signals for –COOH (aspartate and glutamate), and from 1100 to 1000 cm^−1^, the signal for –CH_3_ (alanine, valine, leucine and methionine) are seen [21].

Analyses of functionalized wool fibres by ATR-FTIR spectroscopy revealed that no precise detection of chitosan polysaccharide was possible on the wool fiber surface, most likely due to overlapping of the signals of chitosan and wool. Similar was found in a study of chitosan grafting on wool using anhydrides [5]. In this case, another method of analysis is more appropriate to prove functionalization. For this purpose, potentiometric titration was used, as discussed below.

#### 3.1.2. Functional Groups Determination

The results of the potentiometric titration are shown in Figure 3. For wool, the two types of ionized groups are the side-chain NH_3_^+^ and side-chain COO^−^ groups. For both the amino group and carboxylic acid, the protonation behaviour is a pH-dependent process, so the relative amount of different protonated/deprotonated groups vary as a function of pH. Therefore, the total charge on the wool as protein macromolecules also varies with the pH [25].

The wool titration curve (Figure 2) is characterized by a wide isoelectric range in the interval of pH ≈ 4.5 to 9 (the pH range in which the wool fiber remains electrically neutral).

The mobility of the protein segments within the swollen fibers is limited in this range by the solid nature of the proteins and the cross-covalent bonding, so that a high electrical potential accumulates in the fiber, which limits the penetration of additional hydrogen ions [25]. Lewis and Rippon [26] found that the isoelectric range of wool is in the pH range between 4 and 10. The broad isoelectric point was explained by the assumption that the adsorption of one proton generates a positive charge within the fiber, which has a repulsive potential that reduces the probability of adsorption of a second proton. However, the positive charge generated in the fiber attracts a counter ion into the internal aqueous solution within the fiber and hence the net charge of the wool does not alter. The system is in fact electrically buffered, resulting in the broad part of the titration curve that is not caused by changes in the dissociation constants of the constituent amino acids but rather by changes in the charge conceded by the fiber [25].

Figure 3 shows that the reference wool contains a positive charge (mmol/kg) in the pH range below 4.5 and was not charged in the range between pH = 4.5 and 9 (isoelectric range). At the level of the anionic or cationic plateau, a small amount of ionized amino and carboxyl groups can be seen, both of which are present as end groups in the wool. Above pH = 9, the proportion of negatively charged carboxyl groups is shown in a net as a smaller amount, whilst below pH = 3.5, the equilibrium amount of protonated amino groups accounts for 26.7 mmol/kg.

The isoelectric point of chitosan is in the pH range between 7 and 9 [27]. The adsorption of chitosan onto wool causes a shift of the isoelectric point (IEO) to the right (i.e., higher pH values). The physical, as well as the chemical modification of the wool, shifts the isoelectric point to more alkaline, which clearly proves the attachment of chitosan to the wool. It can be seen, that the positive plateau values in charging isotherms are much higher by functionalized wool samples in comparison to pristine wool, which is due to the higher number of protonated amino groups as a result of additional chitosan-bound amino groups. For sample V + H5, 607.7 mmol/kg of the amino groups were determined, while about 180 mmol/kg amino groups were defined for sample V+H7. It is obvious, and somehow surprising, that the chitosan solution coating based on physical attachment (without crosslinker) offers a better accessibility of the amino groups on the wool when using chitosan solutions of pH = 5 than at pH = 7. At pH = 7, chitosan obtains a coiled structure, which is thermodynamically preferable to being adsorbed in higher quantities. However, at pH = 5, chitosan is in a more extended conformation, and can form thin film layers on the wool by the adsorption, which, consequently, offers better accessibility to the amino groups. Obviously, on the base of protonation behaviour of pristine wool, at both pHs, the hydrogen bonds, Van der Waals and hydrophobic interactions are responsible for chitosan adsorption. The latter are more extensive when the chitosan macromolecular solution is at pH = 5. Namely, proteins exhibit greater surface activity closer to their pI (net charge is zero), due to reduced electrostatic repulsion between the uncharged adsorbing molecules (more molecules to bind) and altered protein structure/conformation (changes in the charge of amino acids) [28]. In addition, wool may be more hydrophobic at an experimental pH = 5, as it is closer to pI and; therefore, more hydrophobic interactions are possible. No Coulombic interactions are possible, whereas raw wool is electrically neutral at this point [28].

Comparing these two samples, V + H5 and V + H7, with wool that was modified chemically by the chitosan macromolecular solution (sample G), it is shown that this sample had 290.7 mmol/kg of amino groups, which is about 40% lower than for the sample V + H5, and, at the same time, contains almost twice the amount of amino groups of V + H7.

In general, it was expected that, by chemical grafting, amino groups are lost through chemical linking and thus more non-available for the protonation.

#### 3.1.3. Stability of Chitosan Coatings

The concentration of desorbed chitosan from the wool surface was determined at pH = 3.6, where chitosan, being completely soluble and thus under strict desorption conditions, the complete (fully) chemical desorption may be evaluated. The stability of chitosan based on physical binding versus chemical grafting was tested and compared in this way. Table 2 shows the volume of PES-Na (as titrant) consumed, and the calculated concentration of desorbed chitosan from physically modified wool (i.e., when a 1% (w/V) macromolecular chitosan solution was applied once at pH = 5 (V + H5) and on the other side at pH = 7 (V + H7), as well as for the sample where chitosan was grafted chemically onto the wool surface (G).

It must be pointed out that reference wool was also subjected to desorption (as blank), and no tracking amino groups were present in this kind of desorption bath.

A bit higher concentration of chitosan was desorbed from physically-functionalized wool by chitosan attachment at pH = 5, compared to physically functionalized wool at chitosan attachment at pH = 7.

If these are compared to the initial values (potentiometric titration curves), we can conclude that the final amount of chitosan onto fibers after strict chemical desorption accounts for 387.7 mmol/kg for V + H5, whereas for sample V + H7 120 mmol/kg. In sample G, no desorption occurred, leading to the final amount of amino groups of 290.7 mmol/kg.

As already pointed out, the physical interaction of wool and chitosan was predicted to be mainly due to weak Van der Waals forces and hydrogen bonds, as well as hydrophobic interactions [29]. As a result, the desorption of chitosan from the wool surface is greater when chitosan is attached physically onto wool fibers. For both samples, V + H5 and V + H7, almost the same amount (i.e., around 35%) of chitosan was desorbed, showing the same trend of coating instability.

The chemical attachment of chitosan to the wool surface did not lead to desorption. Chitosan contains amino groups which bind covalently to the carboxyl groups of wool. Consequently, the chemical modification led to a lower final concentration of protonated amino groups, but to a stable coating due to the covalent bonds established among both substrates.

#### 3.1.4. Sorption Experiments Result

The efficiency of Zn(II) sorption onto pristine wool was determined at 1, 2, 3, 5 and 24 h with Zn(II) initial concentrations of 25, 12.5 and 6.25 mg/L at pH = 5 and pH = 7, respectively. Figure 4 shows that, as Zn(II) sorption onto pristine wool is dependent on time, the equilibrium was reached after 5 h, at pH = 7. Ahmed et al. [13] found out that, at pH = 7, only 2 h were needed to reach the equilibrium, whereas it was not reached at pH = 5 even after 5 h. Other studies reported that equilibrium was reached for pristine wool after 6 h [11,14]. The highest efficiency was determined at equilibrium time of 10 h at pH = 6, reported by the same authors.

The general result to be pointed out is that, at pH = 7, the equilibrium was reached in 5 h with lower initial concentration of 6.25 mg/L, and the efficiency was close to 100%.

If the initial Zn(II) concentration was low, the efficiency was greater compared to higher initial concentrations of Zn(II). At 25 mg/L of Zn(II) initial concentration, 34.1% removal efficiency was reached at pH = 5 and 25.60% at pH = 7. With 12.5 mg/L of initial concentration, the efficiency increased to 63.8% at pH = 5 and 55.4% at pH = 7. With 6.25 mg/L of initial Zn(II) concentration 94.0% and 100%, metal ion removal was reached at both pH values, respectively. All available wool binding sites were filled with Zn(II) ions. The results are consistent with another study of pristine wool where, at low initial concentrations of 1.43 mg/L Zn(II), the efficiency reached 96.4% [14]. At high concentrations of 71.4 mg/L the efficiency was due to wool saturation of only 20.0% [14]. To look in more detail, the amount of Zn(II) sorbed onto wool (mg/g) was determined, and it is shown in Figure 5.

About 10% better efficiency for Zn removal is, in general, seen at pH = 5 in comparison to pH = 7, independent of the initial Zn(II) concentration. From the potentiometric titration curves for pristine wool, it is clearly seen that wool has no charge in this pH interval. All of the present amino groups are; thus, in deprotonated form, and offer a binding site for metal chelation. It was proposed that simultaneously complexing reactions with NH_2_ and metal could occur [15]. Metals form complexes with free electron pairs of amine groups, as presented in Figure 6.

In addition, carboxylic acids are present in all samples in small amounts (as discussed in the table below), allowing the formation of zinc carboxylate complexes, where Zn(II) can bind to one or two carboxylic acids with proton release [29]. In previous studies it has been shown that the possibility of Zn(II) complexation with carboxylic acid groups is not only likely by proton exchange but also by interactions with dissociated groups [29].

#### 3.1.5. Wool Physically Functionalized by Chitosan

The capacity (q) of sorption of Zn(II) on wool physically functionalied by chitosan was determined at pH = 7 (Figure 7 and Figure 8) in order to ensure fully the amount of the deprotonated amino groups and protonated carboxylic acids as binding sites for metals (see Figure 3).

The sorption efficiency of Zn(II) by chitosan—physically modified wool increased with time for initial Zn(II) solution concentrations of 6.25, 12.5 and 25 mg/L in equilibrium time between 5 and 24 h. The time required to reach equilibrium was shorter for those functionalized fibers than for the pristine wool. On the surface of the functionalized wool, the concentration of available deprotonated amino groups was higher in comparison with the pristine wool, whereas applying 1% (w/V) macromolecular solution of chitosan to wool (V + H5 and V + H7) introduced additional amino groups. From the potentiometric titration curves, the total amount of protonated amino groups was determined for V + H5 as 607.7 mmol/kg and for V + H7 as 180 mmol/kg. Thus, the wool sample V + H5 had almost triple Zn(II) sorption sites available; as a result, the time required to fill all the vacancies increased, whilst more time is needed for the equilibrium reaction. From Figure 7 it is evident that the maximum sorption capacity of the sorption experiments for both functionalized samples was achieved with the highest initial concentration of Zn(II) solution (25 mg/L). The maximum sorption capacity of V + H5 was as high as 1.52 mg/g after 24 h, and for V + H7 it was 1.4 mg/g. A slightly lower sorption capacity was obtained with a solution of Zn(II) at an initial concentration of 12.5 mg/L (i.e., 1.09 mg/g for V + H5 and 0.91 mg/g for V + H7). By the lowest Zn(II) solutions concentrations, the sorption capacity of the two sorbents was similar, as all available Zn(II) concentrations were potentially occupied fast. The sorption equilibrium amount for V + H7 was; thus, 0.59 mg/g, and 0.63 mg/g for V + H5. From Figure 7 it is seen that, at the highest initial concentration of Zn(II) solution (25 mg/g), efficiencies were determined to be 63.7% for V + H5 and 55.9% for V + H7, respectively, after 24 h. The sorption capacities were 98.2% for V + H5 and 87.9% for the V + H7 at Zn(II) initial concentration of 12.5 mg/g. With the lowest concentration of Zn(II) solution, similar results regarding sorption efficiency were obtained by both sorbents (up to 100%).

If the results of these functionalized samples are compared with those of pristine wool, the sorption capacity increased in the following way: RV < V + H7 < V + H5.

The reason for this was as already pointed in the higher concentration of deprotonated amino groups present for binding with metal ions. Namely, the sorption of Zn(II) was attributed to the chemical bond between Zn(II) and deprotonated amino groups, whereby a mono- or bi-complex could be formed. It is obvious that the sorption of Zn(II) increases with the higher amount of deprotonated amino groups. To a lesser extent, sorption may also be due to diffusion into pores, physical interactions, and ion-dipole interactions. [13]. The carboxylic acid groups must also be considered, which will be discussed in Table 3 (below).

#### 3.1.6. Wool Chemically Functionalized by Chitosan

Figure 9 shows the efficiency of Zn(II) sorption of three concentrations (25, 12.5 and 6.25 mg/L) onto sample G at pH = 7. The results show that the sorption of Zn(II) onto chemically modified wool is a function of time. By an initial concentration of 12.5 and 6.25 mg/L of Zn(II), equilibrium was not reached after 5 h; this was restored between 5 and 24 h. Equilibrium was reached after 5 h with 25 mg/L initial concentration of Zn(II) solution. The reason is the same as for physically modified wool. A sufficiently high Zn(II) concentration enabled a faster binding to the free binding sites of the chemically modified wool. The sorption efficiency (Figure 9) accounted for 39.1% at 25 mg/L concentration, and 76.6% at 12.5 mg/L concentration. With the lowest initial concentration of Zn(II) solution (6.25 mg/L), almost all available Zn(II) ions were consumed, and the sorption efficiency reached 83.7%. The same trend of concentration effect on sorption efficiency was seen for pristine and chitosan—physically modified wool. Efficiency decreased with increasing the initial concentration.

Capacity (*q_e_*) of Zn(II) adsorption on chemically modified wool was determined as seen in Figure 10. The maximum sorption capacity of Zn(II) after 24 h was found to be 0.91 mg/g, at Zn(II) initial concentration of 25 mg/L as well as 12.5 mg/L. For the lowest initial Zn(II) concentrations, sorption capacity was accounted to 0.58 mg/g in equilibrium.

If the results are compared with those of pristine wool, the sorption capacity increased, but it decreased compared with samples functionalized by chitosan using physical modification. In comparison to sample V + H5, this could be contributed to lower concentration of deprotonated amino groups due to wool crosslinking with chitosan. Therefore, sorption of Zn(II) ions was hindered due to less available amino groups. In comparison to sample V + H7, one would expect better results for G due to the higher amount of deprotonated amino groups in this sample.

This clearly suggests that, besides amino groups, other parameters, such as hydrophilicity, hydrophobicity, fibers‘ morphology and structure, may also play some role for metal sorption ability. During crosslinking with chitosan, the wool structure can also be modified to obtain a denser packaging structure, which may impede sorption capacity; therefore, the penetration of Zn(II) at these binding sites was more difficult, which could outcome from steric effects. The main mechanism of the Zn(II) sorption was again attributed to the formation of a chemical bond, a mono- or bi-complex, between Zn(II) and the amino groups. Sorption may be also done with carboxylic acids and, to some extent, also to metal binding with OH groups. It has been shown by Boguta et al. that Zn(II) also bound to OH groups which was proved by FTIR spectra [29]. Thus, physical interactions and ion-dipole interactions are also involved in sorption mechanisms.

The influence of functional groups was further clarified in order to get more insight into the mechanism of Zn binding to the wool. From potentiometric titration, the number of protonated amino groups, as well as deprotonated carboxylic groups, was determined, and are shown in Table 3. The first was calculated from the positive plateau levels of charging isotherms, whilst the second ones from the negative plateau values. The initial amount, with no desorption taken into account, is given in Table 3. It has been found previously that, at pH = 7 where chitosan is not soluble, the desorption from fibers may not occur, or only to a negligible extent [17]. On the contrary, the desorption results above clearly pointed out the strict chemical desorption at pH = 3.6, where chitosan is fully soluble and; thus, easier to remove from the fiber’s surface [30].

Since sorption experiments were done at pH = 7, all of amino groups for the samples RV, V + H5 and V + H7 were in deprotonated form (see Figure 3) and available for reaction with Zn(II). A total of 50 mmol/kg of amino groups on sample G were still in protonated form, which gives in total 290.0 mmol/kg of deprotonated available amino groups for chelation reaction (see Figure 3). Carboxyl groups were in COOH form for all samples. The correlation between the sorption amount of Zn(II) on amino -NH_2_ and carboxylic acids -COOH is shown in Figure 11.

Figure 11 displays the correlation coefficients. Correlation coefficients are used in statistics to estimate how strong the relationship between two variables is. Here, the correlation is given between amino groups, as well as carboxylic acids and sorption affinity, respectively. In both cases magnitudes are around 0.5, indicating variables which can be considered moderately correlated. For amino groups positive correlation is pointed out, meanwhile, for carboxylic acids, negative correlation is seen, as presented in Figure 6. This may be helpful for explanation of the binding mechanisms of Zn(II) onto wool fibers. An increase in the amount of carboxylic acids leads to a decrease in the value of Zn(II) sorption capacity in mg/g. The latter is anticipated, whilst all groups are at pH of sorption experiment in the protonated form, which cannot act efficiently as a binding site for metals. Oppositely, metals rather bind with COO−. It has been publicized that bidentate bridging coordination through COO^−^ can be a leading mechanism for Zn binding. It has been shown earlier that not all COOH groups are convoluted in Zn(II) binding, which may be due to steric effects [29].

One must also consider the binding of the metal by OH groups, while it increases due to the attendance of hydration water and the formation of aqua complexes. The hydroxyl groups were increased with the introduction of chitosan in wool [29]. In addition, structural parameters (pore size, packing density, degree of the amorphous phase), which are involved in the sorption capacity and binding mechanism, must also be taken into account.

However, the best sorbent was identified by wool physically modified by applying 1% (w/V) macromolecular chitosan solution at pH = 5, where by far the most of the binding sites in the net are obtainable. Additionally, the highest amount of deprotonated amino groups are available for this sample. So, this pointed out that amino groups are the driving force for fast and efficient zinc chelation.

#### 3.1.7. Sorption Kinetics

Sorption kinetics were investigated for the pseudo-first and pseudo-second orders. The calculations were performed over a time range of (0–24) h, with an initial concentration of Zn(II) solutions of 12.5 mg/L. Table 4 shows the calculated pseudo-second order coefficients, and Figure 12 shows that experimentally determined data of unmodified and modified wool fit well to pseudo-second order kinetics.

The pseudo-first-order equation did not fit within the studied time range. The sorption capacity was significantly different from the experimentally determined values. The reason was probably the time delay (retardation phase), which is due to the cross-linking of chitosan with wool fibers and; thus, to the external resistance at the surface of the wool. The best fit of the time range was obtained with the pseudo-second order equation, as shown in Figure 12. The fit coefficients were high (R^2^ > 0.998), and most of the calculated sorption capacities were determined experimentally. A slightly larger deviation was achieved with physically modified wool, V + H5 and V + H7, where the experimentally determined values of sorption capacity were 1.52 and 1.40 mg/g.

The kinetics of Zn(II) sorption onto wool have not yet been investigated, so the results were compared with the kinetics of sorption of a wider range of metals on wool, and the kinetics of Zn(II) sorption on other biosorbents. Zhang et al. [10] investigated the kinetics of sorption for three metals (Cr(III), Cu(II) and Pb(II)) on wool. Two models were paralleled: Pseudo-first and Pseudo-second orders. Whatever metal was selected the fit was toward the pseudo-second order (R^2^ > 0.996). Mishra et al. [19] checked the kinetics of Zn(II) sorption to eucalyptus leaf powder. The sorption of Zn(II) corresponded to the pseudo-second order kinetics, with R^2^ = 0.988 [19].

The kinetics of Zn(II) sorption on wool and other sorbents corresponds to a pseudo-second order, indicating that sorption on the surface of the sorbent proceeded primarily through chemical forces rather than physical forces. The assumption is; thus, confirmed that the formation of complexes between Zn(II) and deprotonated amino groups is predominant in the sorption mechanism.

#### 3.1.8. Sorption Isotherms

The parameters of the extrapolated Langmuir model data are shown in Table 5. By comparing the models, it was found that the experimental data fit the Langmuir model with a coefficient of fit higher than 0.9860. The sorption of Zn(II) took place at specific sites on the surface of pristine and modified wool. We also confirmed that sorption proceeds via a chemical bond. The calculated values of the maximum sorption capacities were like those determined experimentally. Even according to the calculations, wool physically modified by applying 1% (w/V) macromolecular chitosan solution at pH = 5 and sorption experiments at pH = 7, proved to be the sorbent with the highest sorption capacity (1.52 mg/g) and the lowest for pristine wool, with sorption experiments at pH = 7 (0.62 mg/g).

Zhang et al. [3] designed sorption isotherms for the sorption of Pb(II) on pristine wool, dog and human hairs and chicken feathers. The calculated values corresponded to the Langmuir model (R^2^> 0.9990). Vaishnav et al. [31] verified isotherms of Zn(II) sorption models on charcoal obtained from Calotropis procera leaves. A similar fit to the Freundlich and Langmuir models was achieved, with a slightly higher coefficient (R^2^ = 0.997) for the latter. Additionally, Pranata Putra et al. [32] verified isothermal models of Zn(II) biosorption to different biosorbents. The agreement between Langmuir’s and Freundlich’s model was checked; the data fit better to the latter. Nasernejad et al. [33] also checked models for the sorption of Zn(II) on carrot residues. The agreement with Freundlich’s model was better. Mishra et al. [19] used eucalyptus leaf powder to remove Zn(II). He achieved a better agreement with the Langmuir model (R^2^ = 0.995).

In general, it can be summarized that Zn(II) sorption took place on the wool by suggesting that chemical bonds were formed mainly between Zn(II) and the chitosan amine groups. The results consolidate the potentiometric measurements to some extent. Experimental data do not suit the Freundlich model, while *R*^2^ were much lower (from 0.5708 to 0.6415). Due to the Langmuir model fit, it is assumed that, at maximum coverage, there is a mainly monomolecular layer of Zn(II) on the surface, and all the binding sites were energetically equivalent. The latter may again suggest that available functional groups such amino groups and OH groups are responsible for chemical (NH_2_) and physical (OH groups) binding of metal onto wool.

## 4. Thermodynamic Study

The experimental fit of thermodynamic parameters is shown in Figure 13. The calculated thermodynamic parameters are shown in Table 6.

The following thermodynamic processes reaffirm the chemical binding of Zn(II). The negative calculated Gibbs free energy (∆G⁰) values indicate that the process was spontaneous. Negative values of enthalpy (∆H⁰) determined the exothermic character of the process, and demonstrated chemisorption with a value higher than 20 kJ/mol. Values lower than 20 kJ/mol may have been due to a poorer fit of the experimental data, or some sorption by diffusion into pores, physical interactions, or ion-dipole interactions [34]. Due to the Langmuir fit, the latter is more explainable. The entropy values (∆S⁰) were negative, which means that the solid-liquid phase (wool–metal solution) was in a regular state. Results are reliable with the results of another study for chitosan sorbent, alone or with some other polymers‘ combination, which showed spontaneous process ∆*G⁰* < 0 [11,12]. Zhang et al. [11] identified the spontaneous sorption process (∆G⁰ < 0) for chicken feathers, human hair, dog hair and pristine wool as a sorbent for Pb(II) removal. Even Dakiky et al. [12] determined a negative Gibbs free energy for sorption of Cr(VI) onto wool. ∆*S⁰* was negative, suggesting that the randomness at the wool–solution interface decreases during the Zn(II) adsorption. Results of ∆*S⁰* were similar for all functionalized wool samples in comparison with pristine wool.

Thermodynamic analysis showed that the Zn(II) sorption was spontaneous and exothermic in nature and followed a chemical sorption mechanism.

## 5. Conclusions

The wool industry produces large quantities of waste wool, which is an ecological surplus and a polluter. Wool is a renewable and cost-effective material that can be used to sorb a wide range of metals. The modification of wool causes structural changes and influences the sorption capacity. The previously cleaned wool was modified physically by applying a 1% (w/v) macromolecular chitosan solution at pH = 5 and pH = 7, and modified chemically by applying a 1% (w/v) macromolecular chitosan solution at pH = 5. The chemical modification was carried out via an amide bond, with the addition of EDC and NHS crosslinkers in a ratio of 2:1.

The presence of functional groups of untouched and chemically and physically modified wool was not verified by FT–IR spectra, because the peaks in the spectra were superimposed. Therefore, a potentiometric titration of the sorbents was performed, to discuss the sorption capacity as a function of functional ionized groups.

The adsorbent with the highest concentration of protonated amino groups (607.7 mmol/kg) and the highest sorption capacity (1.52 mg/g) was obtained with wool, physically modified by applying a 1% (m/V) macromolecular chitosan solution at pH = 5. The contact time required for the adsorption of Zn(II) on wool was 5 h for modified and non-modified fibers. At the initial concentration of 12.5 mg/g, a contact time of 24 h, pH = 5 and T = 25 °C, the efficiency was 98.19%. The sorption of Zn(II) with natural and all modified wool corresponded to a kinetics of pseudo-secondary order (*R^2^* > 0.9884). The Langmuir model (*R^2^* > 0.9866) proved to be more suitable compared to the Freundlich model. The Zn(II) sorption process was spontaneous (∆*G* < 0) and exothermic (∆*H* < 0). The analysis of the thermodynamic parameters showed that the Zn sorption was spontaneous and exothermic, and followed a chemical sorption mechanism.

It should be noted that functionalization of the wool with chitosan was useful, while the quantity of sorbed Zn(II) increased. With some additional refinements (e.g., applying more layers of chitosan on the wool), even better results can be achieved.

It is of great interest to develop the technology of wool modification to obtain an efficient suspendable sorbent for metals. It is economical and harmless to the environment. Finally, it is biodegradable and the design is often easy to develop. This functionalized wool can be useful for removing Zn(II) from wastewater and thick sludge.

## Figures and Tables

**Figure 1 materials-13-03208-f001:**
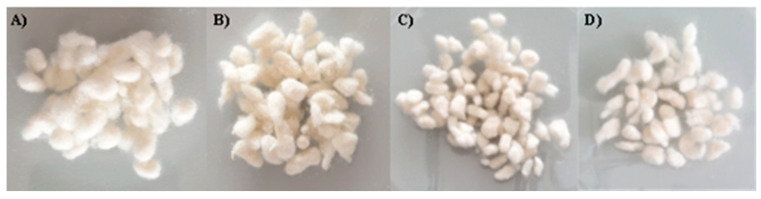
Photos of materials: (**A**) =RV, (**B**) = V + H5, (**C**) = V + H7 and (**D**) = G.

**Figure 2 materials-13-03208-f002:**
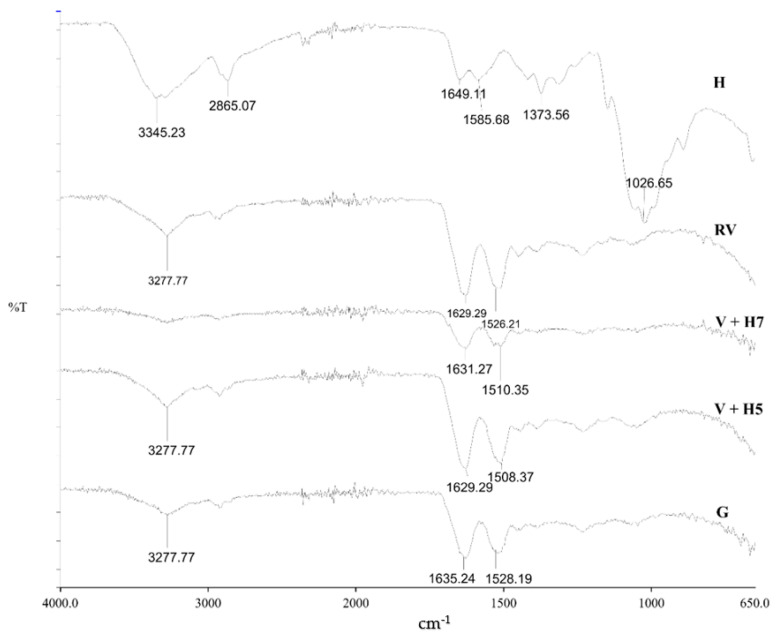
Fourier-transform infrared spectroscopy (FT-IR) spectra of wool (RV), chitosan in granulates (H), physically modified wool with 1% (m/V) chitosan at pH = 5 (V + H5) and pH = 7 (V + H7), and the grafting procedure (G).

**Figure 3 materials-13-03208-f003:**
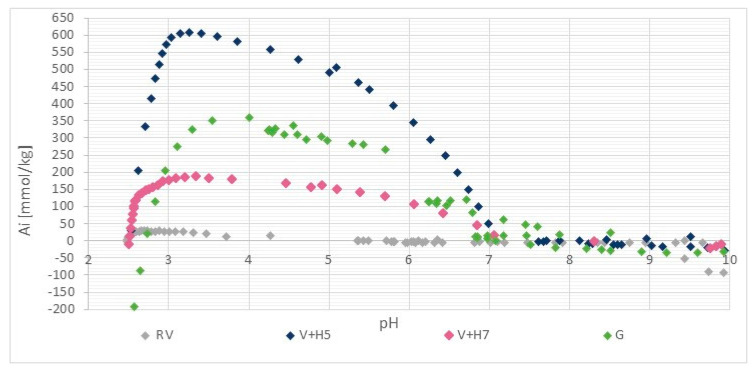
Potentiometric titration curve of pristine wool (RV), physically modified wool at pH = 5 (V + H5) and pH = 7 (V + H7) and chemically modified wool (G). Ai means the amount of charged groups (mmol/kg).

**Figure 4 materials-13-03208-f004:**
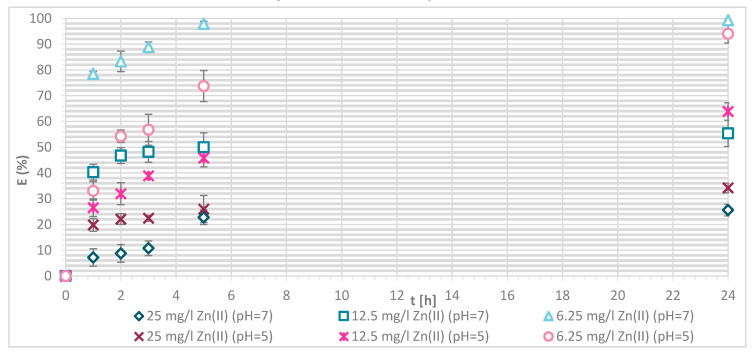
Sorption of Zn(II) on pristine wool (RV) as a function of time and pH (i.e., at pH = 5 and 7, respectively.

**Figure 5 materials-13-03208-f005:**
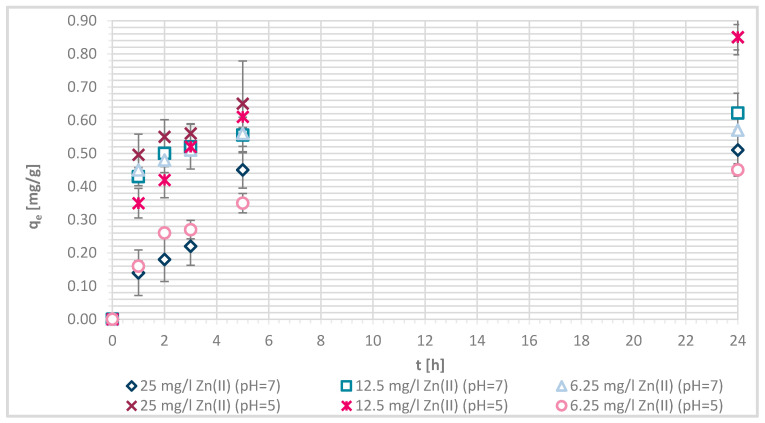
Sorption amount of Zn(II) onto pristine wool (RV) as a function of time.

**Figure 6 materials-13-03208-f006:**
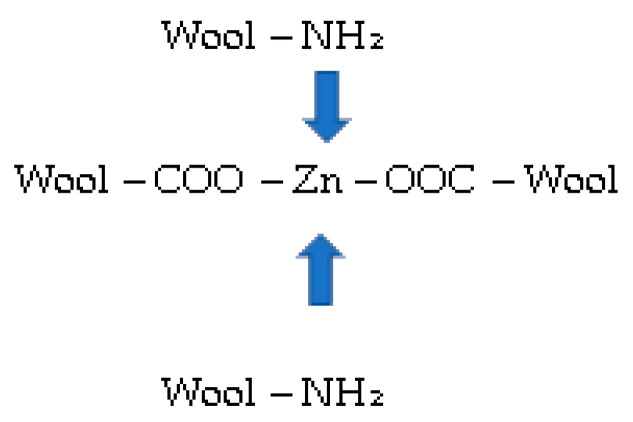
Complexes with metal (Zn) and NH_2_ groups.

**Figure 7 materials-13-03208-f007:**
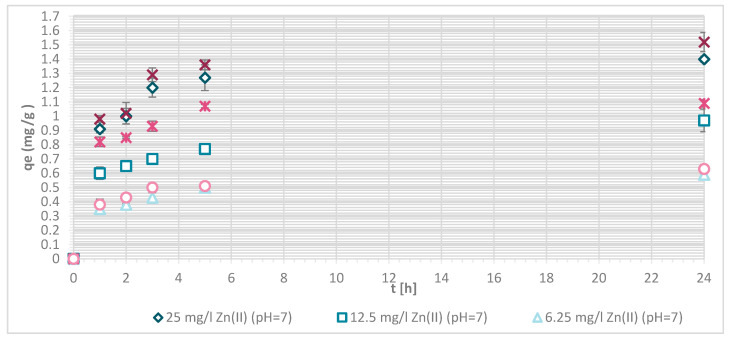
Sorption capacity of Zn(II) on physically modified wool as a function of time, for samples V + H5 and V + H7.

**Figure 8 materials-13-03208-f008:**
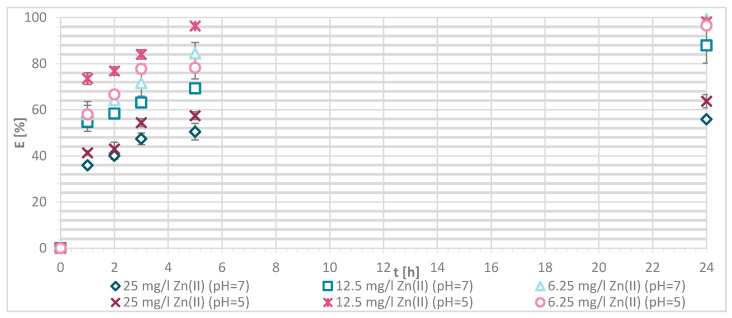
Removal efficiency (E (%)) on physically modified wool as a function of time, for samples V + H5 and V + H7.

**Figure 9 materials-13-03208-f009:**
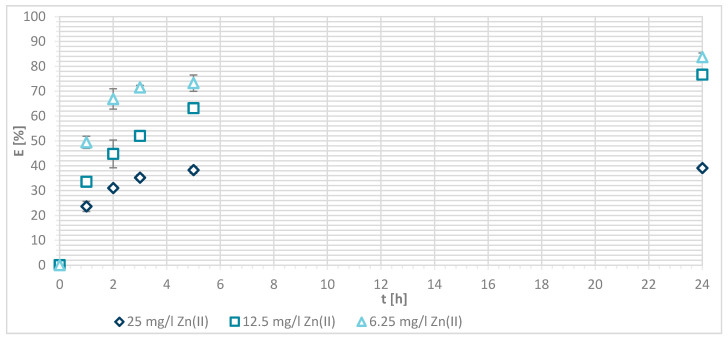
Sorption efficiency of Zn(II) (G) on chemically modified wool as a function of time.

**Figure 10 materials-13-03208-f010:**
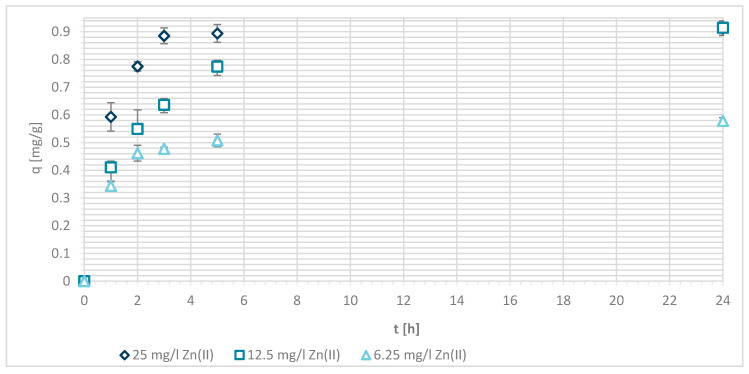
Sorption capacity of Zn(II) (q_e_) on chemically modified wool (G) as a function of time.

**Figure 11 materials-13-03208-f011:**
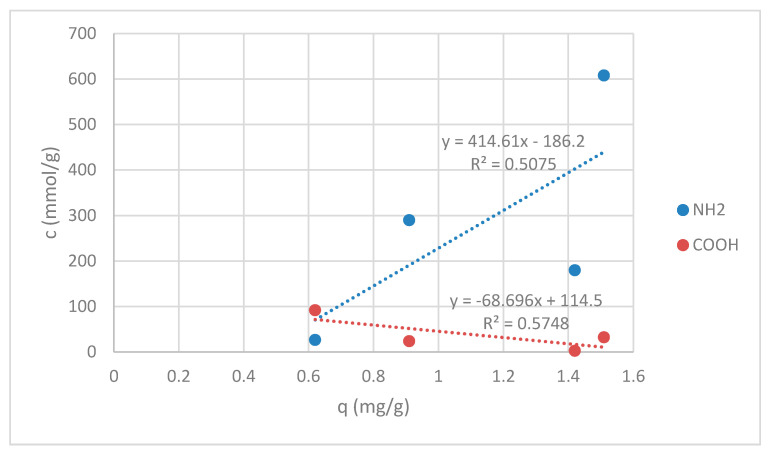
Correlation graph between the sorption amount of Zn(II) and amino -NH_2_ and carboxylic acids -COOH.

**Figure 12 materials-13-03208-f012:**
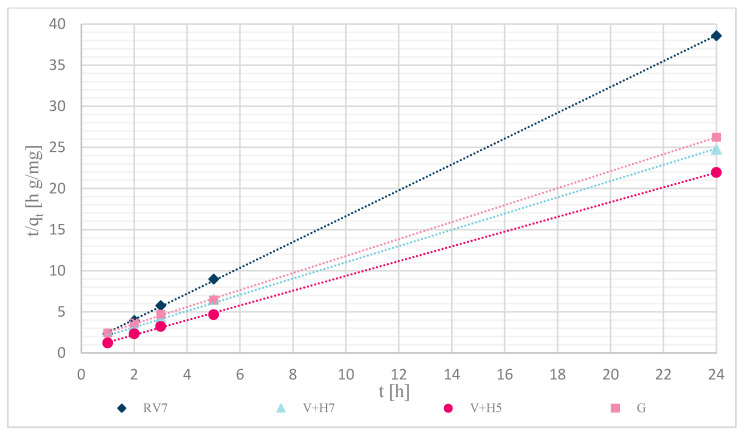
Kinetics of pseudo-second order reaction for samples: RV7, V + H7, V + H5 and G.

**Figure 13 materials-13-03208-f013:**
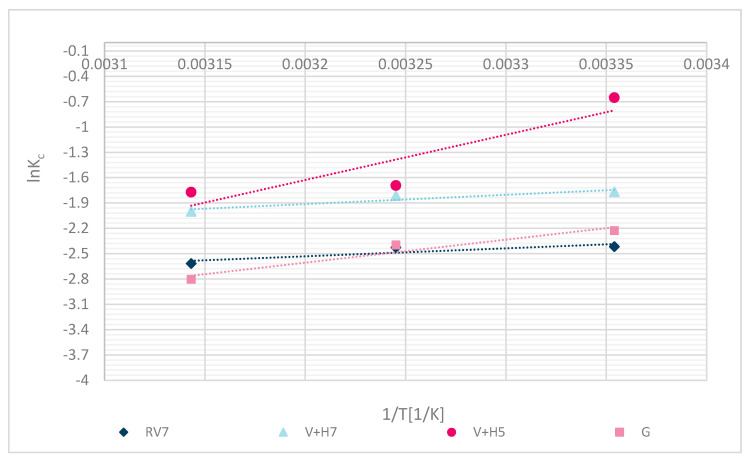
Thermodynamic sorption for different samples RV, V + H5, V + H7 and G.

**Table 1 materials-13-03208-t001:** List of wool samples.

Sample Notation	Description of Sample
RV	Pristine wool
V+H5	Wool modified physically with chitosan macromolecular solution at pH = 5
V + H7	Wool modified physically with chitosan macromolecular solution at pH = 7
G	Chemical modification of wool using the grafting procedure

**Table 2 materials-13-03208-t002:** Results of polyelectrolyte titration: Desorbed chitosan from modified wool (titrated by 0.1 mol/L PES-Na). and calculated amino groups (*c*_c_) in desorption baths. The % of desorption was calculated to the initial amount of amino groups in specific samples, as determined by potentiometric titration.

	Volume of Titrant V_PES-Na_ (mL)	Calculated Amino Groups c_c_ (mmol/kg)	Desorption (%)
**V + H5**	1.1	220	36.2
**V + H7**	0.3	60	33.3
**G**	0	0	0

**Table 3 materials-13-03208-t003:** The number of protonated amino groups, as well as deprotonated carboxylic groups, as determined by potentiometric titration.

Groups	Protonated Amino Groups NH_3_^+^ (mmol/kg)	Deprotonated Carboxyl Groups COO^−^ (mmol/kg)
**RV**	26.7	92.2
**V + H5**	607.7	32.6
**V + H7**	180.0	3
**G**	290.0	24.0

**Table 4 materials-13-03208-t004:** Constants for pseudo-second order reaction.

Sorbent	Equilibrium Concentration of Adsorbed Ions q_e_ (mg/g)	Pseudisecond Order Constant k_2_ (g /(mg h))	Determination Coefficient R^2^
**RV**	0.62	2.63	0.9999
**V + H5**	1.09	1.98	0.9997
**V + H7**	1.01	0.84	0.9987
**G**	0.91	0.72	0.9998

**Table 5 materials-13-03208-t005:** Determination of experimental data according to the Langmuir equation.

Sorbent	Maximum Experimental Sorption Capacity q_max,eksp_ (mg/g)	Maximum Sorption Capacity q_max_ (mg/g)	Langmuir Constant K_L_ (l/mg)	Determination Coefficient R^2^
**RV**	0.62	0.66	6.14	0.998
**V + H5**	1.52	1.53	2.84	0.9866
**V + H7**	1.4	1.42	4.19	0.9971
**G**	0.91	0.94	2.41	0.9983

**Table 6 materials-13-03208-t006:** Parameters for Zn(II) sorption.

Sorbent	Gibbs free energy ∆G⁰ (25 °C) [J/mol]	Enthalpy ∆H⁰ [J/mol]	Entropy ∆S⁰ [J/(mol K)]	Determination CoefficientR^2^
**RV**	−4500.0	−7831.7	−46.1	0.7712
**V + H5**	−2589.4	−44,553.9	−156.1	0.8174
**V + H7**	−3553.8	−9087.7	−45.0	0.8786
**G**	−2183.3	−22,631.2	−94.1	0.9378

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
