# Peer review of "Functionalized Wool as an Efficient and Sustainable Adsorbent for Removal of Zn(II) from an Aqueous Solution"

_materials, 2020, doi:10.3390/ma13143208_

Round 1

Reviewer 1 Report

Dear Authors,

Please find attached my comments on your article. It could be interesting but major revision should be made and some parts of manuscript needs to be explained or improved.

Kind regards

Reviewer

Reviewer 2 Report

Review Report for Metals manuscript n. 855902

The topic of the paper is interesting and match the scopes of the journal. The sorbent used in the study is a natural fiber that nowadays is only partially used for clothes or other standard uses, and most of the wool produced worldwide is simply thrown away or burnt. This alternative use is certainly of interest to wastewater treatments and is a perfect example of circular economy.

The Materials& Methods section is well written and explained. Nevertheless, you prepared Zn-containing solutions with initial concentrations of 25 mg/L, 12.5 mg/L and 6.25 mg/L for sorption studies, and the pH values that you investigated were 5 and 7.

After the pH adjustment, have you measured again the initial concentration of Zn? As Zn usually precipitates partially with those pHs, so the results of the adsorption studies could be rather different... Please clarify that.

The present paper investigated an interesting process for the production of Sb, and, in my opinion, it deserves to be published in Meterials.

I also recommend to review the language, there are some errors like "The influence was investigated of contact time,.."

Round 2

Reviewer 1 Report

Dear Authors,

I am glad to see your improved manuscript. Now I recommend to accept it. However some minor corrections have to be introduced. Please find below my comments.
